# Structural and functional analysis of human pannexin 2 channel

Zhihui He[1,2,11], Yonghui Zhao[3,4,11], Michael J. Rau[5], James A. J. Fitzpatrick[1,5,6,7], Rajan Sah[2,8], Hongzhen Hu[3,4] ✉ & Peng Yuan[1,2,9,10] ✉

The pannexin 2 channel (PANX2) participates in multiple physiological processes including skin homeostasis, neuronal development, and ischemia-induced brain injury. However, the molecular basis of PANX2 channel function remains largely unknown. Here, we present a cryo-electron microscopy structure of human PANX2, which reveals pore properties contrasting with those of the intensely studied paralog PANX1. The extracellular selectivity filter, defined by a ring of basic residues, more closely resembles that of the distantly related volume-regulated anion channel (VRAC) LRRC8A, rather than PANX1. Furthermore, we show that PANX2 displays a similar anion permeability sequence as VRAC, and that PANX2 channel activity is inhibited by a commonly used VRAC inhibitor, DCPIB. Thus, the shared channel properties between PANX2 and VRAC may complicate dissection of their cellular functions through pharmacological manipulation. Collectively, our structural and functional analysis provides a framework for development of PANX2-specific reagents that are needed for better understanding of channel physiology and pathophysiology.

The pannexin family of ion channels comprises three paralogs, PANX1-3, which differ in tissue expression and physiological functions[1,2]. PANX1, a plasma membrane ATP-release channel, is involved in numerous physiological processes associated with purinergic signaling[3,4]. Missense mutations in PANX1 have been linked to multisystem dysfunction including abnormal oocyte development[5,6], and thus PANX1 represents a promising therapeutic target and has been intensely studied since its molecular identification. With recent technical breakthroughs in single-particle cryo-electron microscopy (cryo-EM), multiple independent laboratories have elucidated cryo-EM structures of PANX1, revealing an unprecedented heptameric channel assembly and providing molecular insights into its permeation and gating mechanisms[7-14].

In contrast to PANX1, much less is known about the other two family members, PANX2 and PANX3, with regard to their biophysical properties and physiological functions, partially owing to a lack of selective pharmacological reagents targeting these channels. PANX2 was initially thought to be expressed selectively in the central nervous system (CNS)[15,16], but was later shown to have broad expression across multiple tissues including the skin[17-19]. Physiologically, PANX2 has been proposed to contribute to neuronal development, ischemia-induced neurodegeneration, and UV radiation-induced apoptosis of keratinocytes[19-21].

[1]Department of Cell Biology and Physiology, Washington University School of Medicine, Saint Louis, MO, USA. [2]Center for the Investigation of Membrane Excitability Diseases, Washington University School of Medicine, Saint Louis, MO, USA. [3]Department of Anesthesiology, Washington University School of Medicine, Saint Louis, MO, USA. [4]Center for the Study of Itch and Sensory Disorders, Washington University School of Medicine, Saint Louis, MO, USA. [5]Washington University Center for Cellular Imaging, Washington University School of Medicine, Saint Louis, MO, USA. [6]Department of Neuroscience, Washington University School of Medicine, Saint Louis, MO, USA. [7]Department of Biomedical Engineering, Washington University in Saint Louis, Saint Louis, MO, USA. [8]Department of Internal Medicine, Cardiovascular Division, Washington University School of Medicine, Saint Louis, MO, USA. [9]Department of Pharmacological Sciences, Icahn School of Medicine at Mount Sinai, New York, NY, USA. [10]Department of Neuroscience, Icahn School of Medicine at Mount Sinai, New York, NY, USA. [11]These authors contributed equally: Zhihui He, Yonghui Zhao. ✉e-mail: hongzhen.hu@wustl.edu; peng.yuan@mssm.edu

The transmembrane domain (TMD) of the pannexin channels (PANX1-3), presumably consisting of four membrane-spanning helices TM1-4, is structurally related to the other "large-pore channels" including connexin[22], innexin[23], volume-regulated anion channels (VRACs) formed by leucine-rich repeat-containing 8 proteins (LRRC8)[24,25], and calcium homeostasis modulators (CALHMs)[26,27]. These large-pore channels differ in ion selectivity and pore properties, and some are permeable to classical ions such as $Na^+$, $K^+$ and $Cl^-$ as well as to larger molecules and metabolites such as ATP, glutamate and cyclic dinucleotides[28–31]. Remarkably, these structurally conserved membrane proteins assemble into distinct oligomeric channels[30]. Connexin hemi-channels and LRRC8 channels are hexameric, and innexin channels are octameric, whereas CALHM channels range from 8-mer to 13-mer[30]. Historically, PANX1 has been long thought to form hexameric channels[29,32]. Until recently, high-resolution cryo-EM structures of PANX1 unambiguously reveal a heptameric channel architecture[7–14]. PANX2, on the other hand, has been suggested to form octameric channels[33], but the low-resolution nature of cross-linking and native gel electrophoresis methods raises uncertainty. It also remains unknown whether PANX1-3 channels, like CALHMs, form distinct oligomeric channel assemblies. Moreover, the common channel architecture, particularly the analogous extracellular constriction and transmembrane pore structure, suggests the possibility of shared pharmacological properties across the extended family of large-pore channels. If true, this would raise important considerations when dissecting the functional roles of individual family members in a cellular context by application of pharmacological agents, which could potentially modulate the activity of multiple distinct channels. For instance, it is known that gap junction connexin channel blockers, such as carbenoxolone (CBX) and flufenamic acid, also inhibit PANX1[34,35].

To advance our molecular understanding of pannexins, here we present a cryo-EM structure of human PANX2. Together with electrophysiological analysis, these results provide previously unknown insights into the structure, function, and pharmacology of PANX2, extend our understanding of the superfamily of large-pore channels, and facilitate development of specific reagents essential for better understanding of PANX2 channel physiology and pathophysiology.

## Results

### Structure determination facilitated by protein engineering

Low levels of expression of the full-length human PANX2 in heterologous systems prevented isolation of channel protein in sufficient quantity for structural studies. To overcome this technical challenge, we engineered a truncated fusion construct, termed PANX2$_{EM}$, consisting of residues 1-372 of human PANX2 connected to a C-terminal thermostabilized BRIL protein (Fig. 1a), which is often used as a fusion partner to facilitate structure determination of integral membrane proteins by X-ray crystallography or cryo-EM[36–38]. The removed C-terminal portion of human PANX2, including residues 373–677, is predicted to be unstructured by AlphaFold[39]. The resulting construct, PANX2$_{EM}$, when heterologously expressed in mammalian cells, showed current–voltage relationship and current density analogous to those of the wild-type full-length channel (Fig. 1b, c), suggesting that PANX2$_{EM}$ maintains functional integrity and that information gained from structural analysis of PANX2$_{EM}$ is physiologically relevant.

We purified the PANX2$_{EM}$ protein to homogeneity, which was subjected to single-particle cryo-EM analysis. The two-dimensional class averages clearly indicated a heptameric channel assembly similar to PANX1[7–9], thus discrediting the previous proposal of an octameric stoichiometry of PANX2[33] (Supplementary Fig. 1). We determined the cryo-EM structure of PANX2$_{EM}$ to an overall resolution of 3.92 Å with applied C7 symmetry (Fig. 1d–g, Supplementary

Fig. 1, Supplementary Table 1). The three-dimensional reconstruction without imposed symmetry constraints (C1 symmetry), albeit at a lower resolution, aligned well with the C7 reconstruction, indicating that PANX2$_{EM}$ indeed formed a symmetric channel assembly. The side-chain densities for the majority of amino acids were well resolved in the cryo-EM density map (Fig. 1d, Supplementary Fig. 2). The AlphaFold model of human PANX2, containing amino acids 1-372, was used as an initial model and placed in the density map to facilitate model building. The final refined atomic model, consisting of residues 35–166, 208–266, and 273–368, fits well into the cryo-EM density with good stereochemistry (Supplementary Fig. 1, Supplementary Table 1) and is further corroborated by the conserved disulfide bonds in the extracellular domain (Fig. 1g, Supplementary Figs. 2 and 3)[7–9]. The N- and C-terminal portions, including residues 1–34 and 369–372, respectively, the intracellular and extracellular loops, consisting of residues 167–207 and 267–272, respectively, and the C-terminally fused BRIL protein were not resolved in the cryo-EM reconstruction.

### Heptameric PANX2 channel

The human PANX2 channel forms a symmetric heptamer with a central ion conduction pore along the seven-fold symmetry axis (Fig. 1d–f). The channel comprises three layers constituted by the extracellular domain (ECD), transmembrane domain (TMD), and intracellular domain (ICD). Each subunit consists of an intracellular N-terminal tail and a C-terminal domain (CTD), a TMD with four membrane-spanning helices TM1-4, and an ECD composed of two extracellular loops, ECL1 and ECL2, connecting TM1-TM2 and TM3-TM4, respectively. On the extracellular side, E1β and E1H from ECL1 and E2β1 and E2β2 from ECL2 assemble into an antiparallel three-stranded β-sheet packing against an α-helix (E1H). The ECD structure is reinforced by two pairs of disulfide bonds, C81-C279 and C99-C259, which crosslink the ECL1 and ECL2 loops (Fig. 1g). The ECD structural unit, including the disulfide linkage, is a conserved feature among the large-pore channels and appears to create a rather rigid extracellular constriction pivotal for ion selectivity and pharmacological inhibition[7–14,24,25]. On the intracellular side, the ICD consists of a compact helical domain, which is constituted by the C-terminal helices CTH1-4 and the intervening helices, CLH1 and CLH2, that connect TM2 and TM3. The subunit structure, inter-subunit arrangement, oligomeric state, and overall channel architecture are analogous to those of PANX1[7–14].

Lipid-like densities are observed at the periphery of the PANX2 channel and at the crevice between adjacent subunits in the transmembrane region (Figs. 1d and 2a). The inter-subunit packing is more relaxed within the membrane than in the extramembrane regions, generating lateral opening, or fenestration, to the lipid bilayer (Fig. 2a). Several lipid molecules snugly intercalate into the subunit interface. These interfacial lipids appear to be obligatory to physically seal the central pore from exposure to the surrounding membrane. The lateral opening, with a decreased magnitude, has also been observed in the PANX1 channel (Fig. 2b)[8], indicating a common structural feature of PANX channels. Lipid penetrating fenestration, often observed in ion channels such as the voltage-gated sodium channels[40], two-pore potassium channels[41], and the calcium uniporters[42], suggests at least two immediate possibilities. First, lipids may directly modulate channel activity such as blocking ion conduction by inserting their hydrophobic acyl chains into the central permeation path[41]. Second, the channel could sense and respond to membrane dynamics, such as elevated membrane tension[43,44], through an intimate association with the bound lipids. Notably, how lipids modulate PANX2 channel activity remains to be investigated. Nonetheless, the large-pore dimension in the transmembrane region would allow pore-residing lipids block conduction, as suggested by cryo-EM structures of related large-pore channels including PANX1 and CALHM channels[14,27].

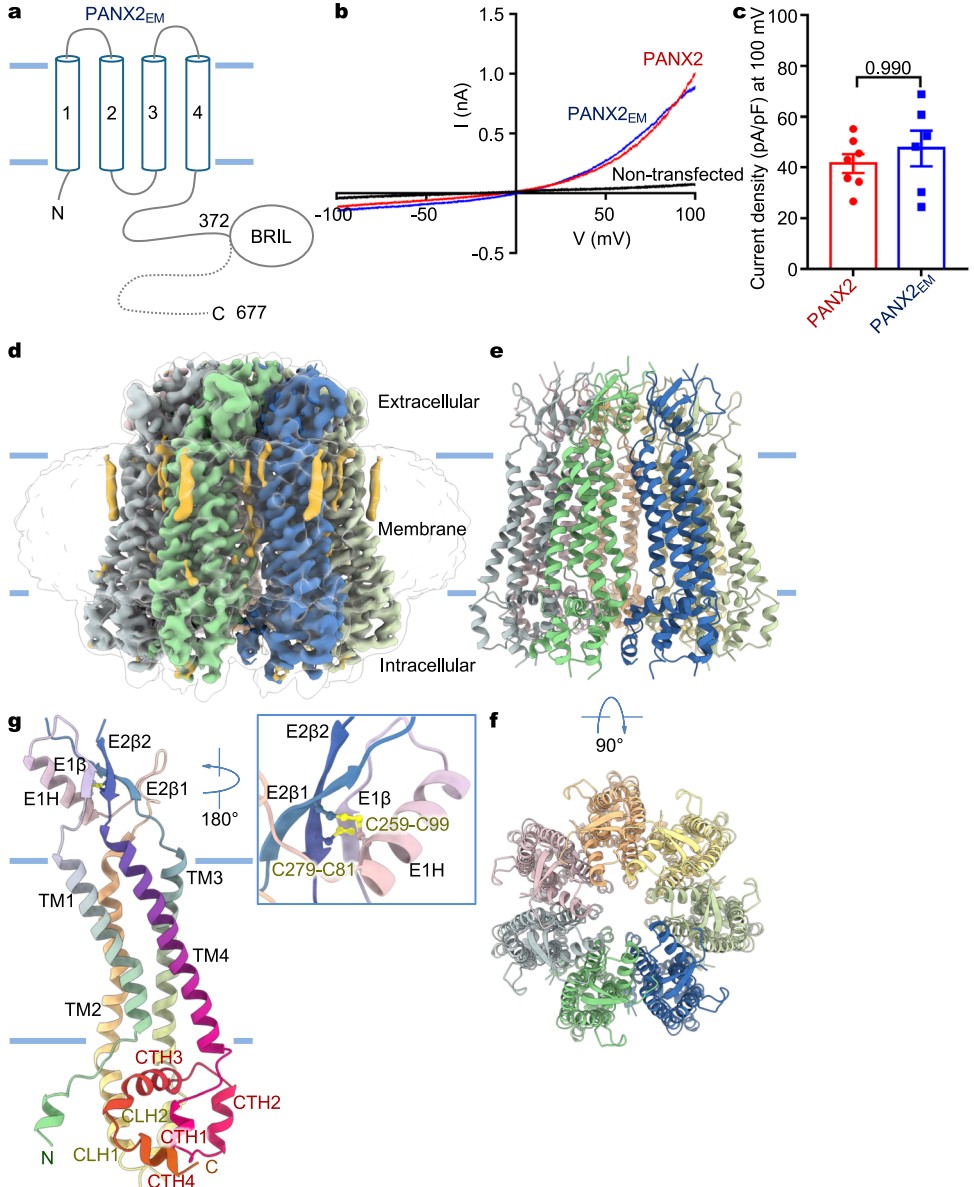

**Fig. 1 | Structure of human PANX2. a** Domain topology of human PANX2. The engineered protein construct PANX2$_{EM}$ for structure determination is indicated. **b** Current–voltage curves for the wild-type full-length human PANX2 and PANX2$_{EM}$ expressed in HEK293T cells. Currents from untransfected cells were used as the negative control. **c** Current density for the full-length and PANX2$_{EM}$. Data are presented as mean ± SEM. *p*-value is indicated (unpaired two-tailed *t*-test, *n* = 7, 6

independent cells for the full length and PANX2$_{EM}$, respectively). **d** Cryo-EM reconstruction of PANX2$_{EM}$ with each subunit uniquely colored. The micelle densities are indicated. Lipid-like densities within the membrane region are colored in orange. **e, f** Structure of human PANX2 in orthogonal views. **g** Structure of a single subunit with secondary structures indicated. The disulfide bonds between conserved cysteine residues in the extracellular domain are highlighted.

## Ion conduction pore

The ion conduction path of PANX2 spans the extracellular, transmembrane, and intracellular regions. On the extracellular side, the seven ECDs assemble into a cap structure, positioning the N-terminal ends of seven E1H helices from the heptameric channel to line the extracellular entrance (Fig. 3a). Each of the seven E1H helices harbors a conserved basic amino acid R89 (Supplementary Fig. 3), and the seven arginine amino acids form a basic ring structure generating an electropositive extracellular constriction, which is additionally contributed by the E1H helix dipole[25]. The hydrated pore diameter at this location is ~6 Å (Fig. 3b), sufficiently wide for passage of hydrated or partially dehydrated ions. In contrast, the equivalent position in PANX1 holds an aromatic amino acid W74, which forms an extracellular selectivity filter (SF) also critical for pharmacological blockade by carbenoxolone (CBX)[7–9,45]. Within the membrane, TM1 and TM2

together line the pore, giving rise to a voluminous vestibule with a minimal diameter of larger than 16 Å.

On the intracellular side, a short α-helix (the N helix, residues 36-40) preceding TM1 constricts the pore with a diameter of ~12 Å, which presumably allows conduction of ions and small molecules (Fig. 3). The PANX2$_{EM}$ construct contains the complete N-terminus, but the first 34 amino acids are not resolved in the cryo-EM density. Thus, the structural and functional role of the extreme N-terminus remains undefined. Nonetheless, the strategic positioning of the N-terminus inside the pore lumen suggests its potential involvement in modulation of the central ion pathway. Notably, the N-terminal segment preceding TM1 appears to form a structurally dynamic element that is critical to regulation of channel activity across the superfamily of large-pore channels[30]. Recent studies of the closely related human PANX1, embedded in lipid nanodiscs, have

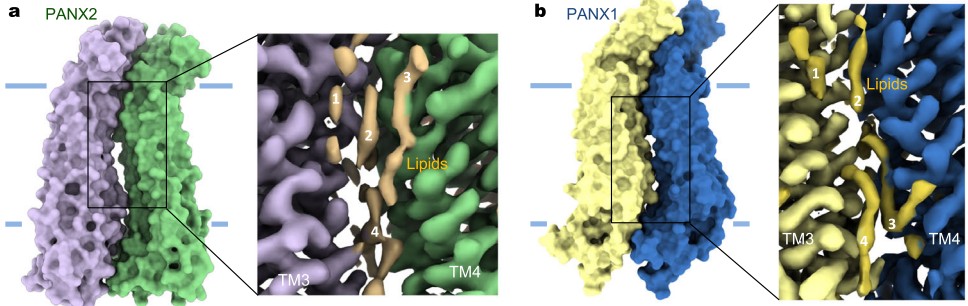

**Fig. 2 | Fenestration in PANX2 and PANX1. a** Lateral opening between two adjacent PANX2 subunits. Lipid-like densities (in orange) are numbered as lipid 1 to 4. **b** Lateral opening between two neighboring PANX1 subunits (PDB: 6UZY). Interfacial lipid densities (in orange) are similarly numbered.

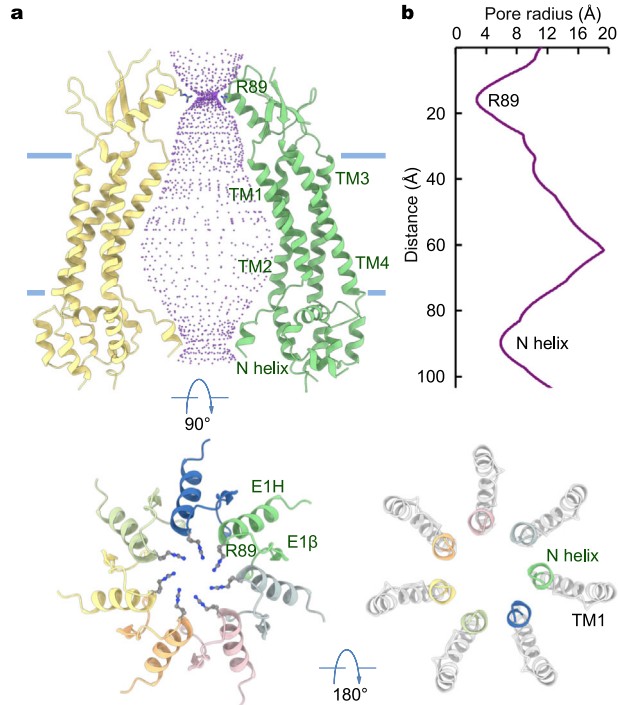

**Fig. 3 | Ion conduction pore of PANX2. a** The pore of PANX2 The extracellular constriction formed by a ring of seven arginine amino acids (R89) from E1H is highlighted. Only two opposing subunits are shown in the top panel for clarity. The extracellular and intracellular constrictions, which are defined by R89 and the N-terminal helices, respectively, are highlighted in the bottom panels. **b** Pore dimension along the conduction path.

demonstrated that deletion of the N-terminus results in a nonfunctional channel and suggested that the movement of the N-terminus accompanied by rearrangement of pore-blocking lipids is obligatory for channel opening[14]. Inspired by our structural observation, we generated a truncation construct of human PANX2 lacking the N-terminal 40 amino acids (PANX2 dN) and found that channel-elicited currents were essentially abolished, though the truncated channel had a comparable level of surface expression as the full-length (Supplementary Fig. 4). This observation, in accordance with the previous finding in PANX1[14], further supports the theme that the structurally dynamic N-terminus is indispensable for channel function. However, how the N-terminal tail contributes to PANX2 channel function has yet to be investigated by further structural and/or molecular dynamics simulation studies.

## Structural comparison with PANX1 and LRRC8A

The structure of each subunit of the PANX2 channel closely resembles that of PANX1 (Fig. 4a), with a root-mean-square deviation (RMSD) of ~1.9 Å for Cα atoms in the transmembrane helices TM1-TM4. However, the heptameric PANX1 and PANX2 channel assemblies deviate from each other to a larger extent, with an RMSD of ~3.7 Å for Cα atoms in all seven TM1-TM4 (Fig. 4b). Structural similarity between PANX1 and PANX2 results in analogous overall pore profiles (Fig. 4c). Nonetheless, in comparison with PANX1, the seven TMDs in PANX2 are positioned further apart, rendering a larger transmembrane vestibule (Fig. 4b, c). In contrast, both the extracellular and intracellular constrictions in PANX2 are narrower than the corresponding regions in PANX1. On the extracellular side, the pore diameters at the R89 ring in PANX2 and the W74 ring in PANX1 are 6 and 9 Å, respectively, which are determined by a combination of the positioning of E1H and the side-chain configurations (Fig. 4b). In the intracellular entrance, the N helix in PANX2, extends towards the central pore axis, further constricting the ion pore. In PANX1, the extracellular constriction is critical for both ion selectivity and pharmacological properties[7–9,45]. In analogy, the corresponding extracellular region in PANX2 may play a similar structural and functional role. Therefore, different amino acid compositions in the presumed extracellular SF suggest distinct ion selectivity and pharmacology of PANX2 from PANX1.

Notably, the ion conduction pore of PANX2 and particularly the ring of basic amino acids lining the extracellular SF are reminiscent of those of the distantly related hexameric LRRC8A channel, despite low sequence identity and distinct subunit stoichiometry[24,25]. The arrangements of the four TM helices are similar in PANX2 and in LRRC8A (Fig. 4d). The heptameric, instead of hexameric, assembly in PANX2 places opposing subunits further away, thus creating a larger central ion conduction pore, especially in the transmembrane region (Fig. 4e). On the extracellular side, in comparison with an opening of 6 Å in diameter of the R89 ring in PANX2, the equivalent R103 ring in LRRC8A has a reduced diameter of ~4 Å.

An extracellular constriction, formed by the N-terminal ends of the E1H helices, is a common structural feature observed amongst these different channels (Fig. 4f–h). Tight intra- and inter-subunit packing interactions in the ECD, augmented by two pair of conserved disulfide bonds, appear to be well suited to generate a rigid structural scaffold to accommodate a fixed extracellular SF. The most critical amino acid, R89, defining the narrowest point in the SF is hosted by an α-helix, rather than by a structurally more dynamic loop as observed in transient receptor potential (TRP) channels[46], further suggesting the inflexible nature of the SF in PANX and LRRC8 channels. Consistent with this notion, all the recently determined PANX1 structures[7–14], either in detergent micelles or lipid environments, essentially have the same SF configuration.

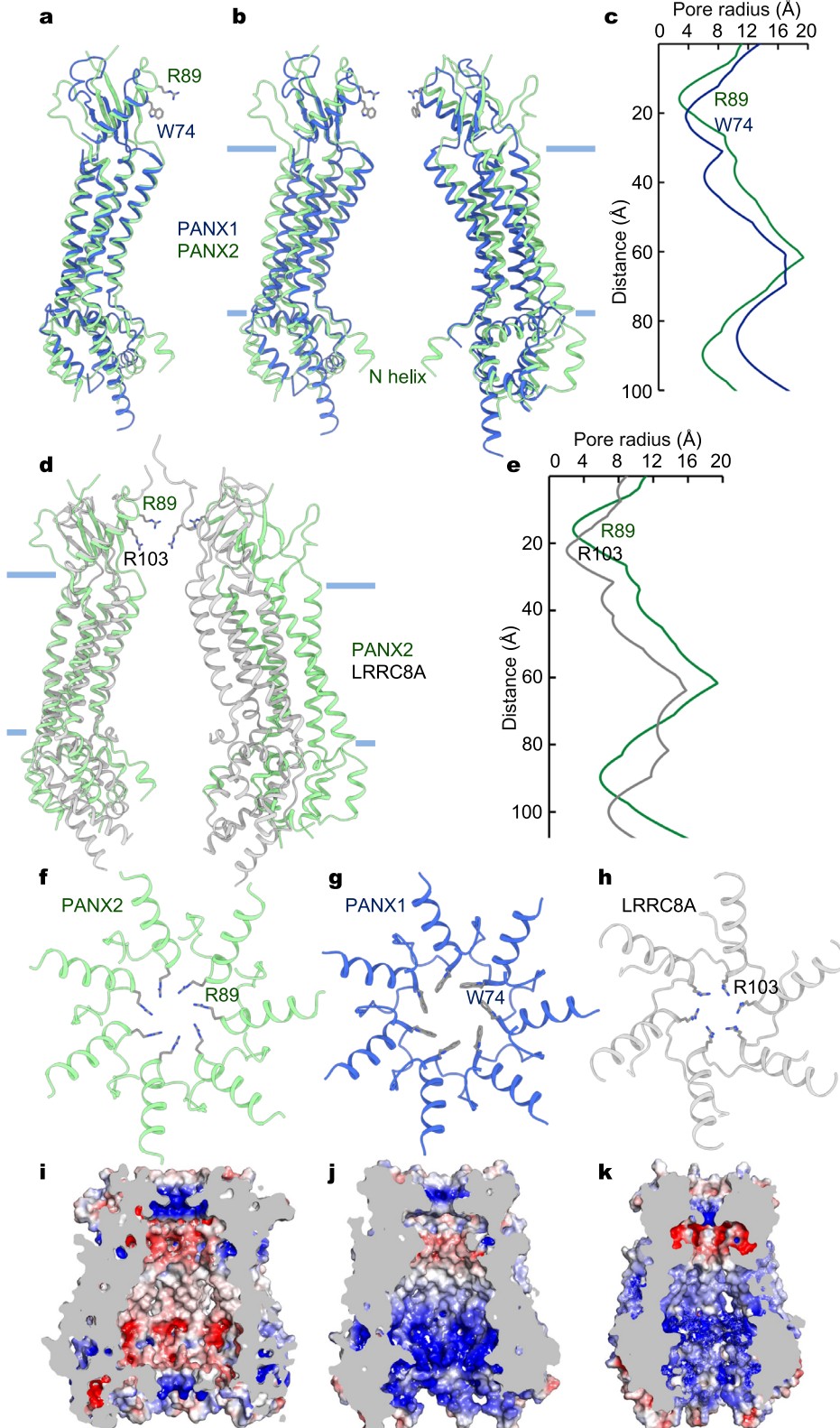

**Fig. 4 | Pore properties of PANX2, PANX1, and LRRC8A channels.**
**a** Superposition of PANX2 (green) and PANX1 (blue) protomers (PDB: 6UZY).
**b** Superposition of the heptameric PANX2 and PANX1 channels. Only two subunits are shown. **c** Pore profiles of PANX2 and PANX1. **d** Overlay of heptameric PANX2 (green) and hexameric LRRC8A (gray, PDB: 6NZW) channels based on superposition of a single protomer from each channel. **e** Pore profiles of PANX2 and LRRC8A. **f**–**h** The putative extracellular selectivity filters of PANX2 (**f**), PANX1 (**g**), and LRRC8A (**h**). **i**–**k** Cutaway views of the central pores of PANX2 (**i**), PANX1 (**j**), and LRRC8A (**k**) colored by surface electrostatic potential (red, −5 kT/e; white, neutral; blue, +5 kT/e).

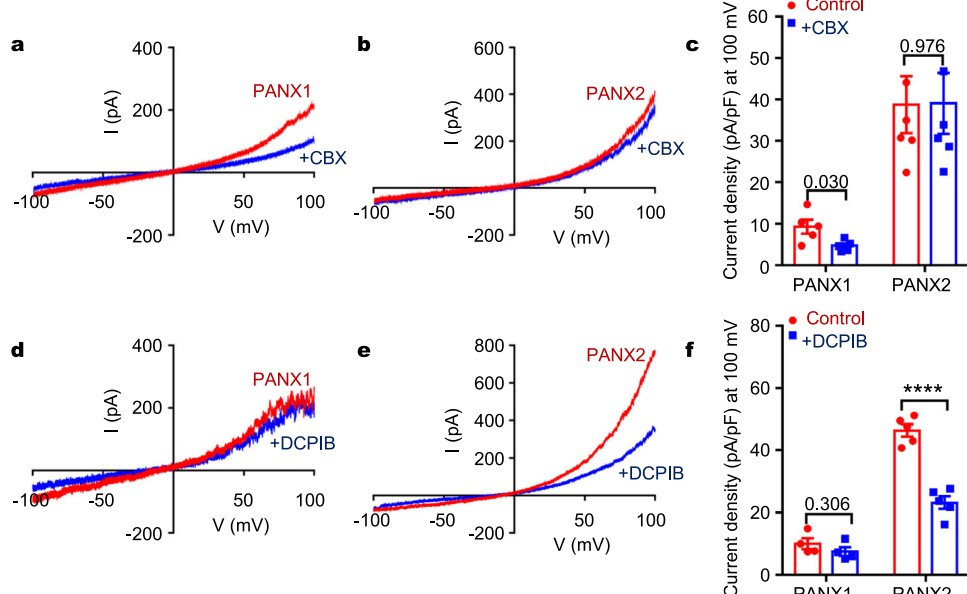

**Fig. 5 | Pharmacology of human PANX2. a, b** Current–voltage relationship of PANX1 (**a**) and PANX2 (**b**) in the absence or presence of 0.1 mM CBX. **c** Current densities of PANX1- and PANX2-mediated whole-cell currents in the absence or presence of 0.1 mM CBX (mean ± SEM, $n = 5$ independent cells, unpaired two-tailed $t$-test). **d, e** Current–voltage relationship of PANX1 (**d**) and PANX2 (**e**) in the absence or presence of 75 µM DCPIB. **f** Current densities of PANX1- and PANX2-mediated whole-cell currents in the absence or presence of 75 µM DCPIB (mean ± SEM, $n = 4$, 5 independent cells for PANX1 and PANX2, respectively, **** indicates $p < 0.0001$, unpaired two-tailed t-test).

## Ion selectivity

The pore lumens in PANX2, PANX1 and LRRC8A display distinct profiles in surface electrostatic potential (Fig. 4i–k). The overall positive electrostatic potential in the pore lumens of PANX1 and LRRC8A supports the anion preference in these two channels. In contrast, the PANX2 pore has a markedly different profile in electrostatic potential (Fig. 4i). Whereas both the extracellular and intracellular entrances exhibit positive electrostatic potential, the transmembrane pore region is largely negative in electrostatic potential.

PANX2 had a slight anion preference, and the permeability ratio of $P_{Cl}/P_{Na}$, as assessed by reversal potential measurements, was less than twofold (Supplementary Fig. 5a). Furthermore, PANX2 was permeable to large anions and cations such as gluconate and N-methyl-D-glucamine (Supplementary Fig. 5b, c). To determine the anion permeability preference of PANX2, we used equimolar Br⁻ or I⁻ as a replacement for Cl⁻ (Supplementary Fig. 5b, c). Interestingly, the PANX2 channel displayed a similar anion permeability sequence of I⁻ > Br⁻ > Cl⁻ as the VRAC channel[47]. The ion selectivity profile of PANX2_EM is essentially identical to that of the full-length PANX2, further confirming that PANX2_EM retains the biophysical properties of the intact channel. Since the R89 ring is a critical component of the SF in PANX2, we reasoned that this basic residue could be a major determinant of the underlying anion selectivity. We substituted R89 with alanine (R89A), tryptophan (R89W), or glutamic acid (R89E) and evaluated these mutant channels by electrophysiology. Notably, all mutant channels displayed significantly reduced currents in whole-cell patch-clamp recordings (Supplementary Fig. 5d, f), though surface expression of these mutant channels is comparable to that of the wild type (Supplementary Fig. 4c, d). Nonetheless, R89W renders the channel essentially non-selective ($P_{Cl}/P_{Na}$ ~ 1.0) (Supplementary Fig. 5e), further supporting that the extracellular basic ring formed by R89 is essential for ion selectivity.

## Pharmacological inhibition of PANX2

Previous studies in PANX1 and LRRC8A channels have revealed that the extracellular SF is critical for not only ion selectivity but also pharmacological inhibition[7–9,24,25,45]. In PANX1, the W74 ring is a major molecular determinant of CBX blockade in that amino acid substitutions at this position diminished or abolished CBX inhibition[7–9]. The equivalent amino acids at this position are Arg and Ile in PANX2 and PANX3, respectively (Supplementary Fig. 3). The wild-type PANX3 channel was insensitive to CBX, but swapping the extracellular ECL1 loop of PANX1 into PANX3 conferred CBX blockade in the chimeric channel[45]. Consistent with these findings, unlike PANX1, PANX2-mediated currents were insensitive to the application of 100 µM CBX (Fig. 5a–c), further highlighting the critical role of W74 in CBX blockade in PANX1. However, introduction of a tryptophan mutation at this position in PANX2 (R89W) did not lead to CBX sensitivity (Supplementary Fig. 6a, c), indicating that additional elements, possibly from neighboring residues in the ECL1 loop, are necessary for CBX interactions. Collectively, PANX1-3 channels may have similar subunit structures and oligomeric assemblies but differ substantially in pharmacology.

The extracellular Arg rings in the heptameric PANX2 (R89) and hexameric LRRC8A (R103) channels are remarkably similar to each other, suggesting the possibility of shared pharmacological properties. DCPIB (4-(2-butyl-6,7-dichloro-2-cyclopentyl-indan-1-on-5-yl) oxobutyric acid), a potent inhibitor for VRAC (or LRRC8), blocks LRRC8A via a "cork in a bottle" mechanism through insertion into the extracellular pore constriction and interaction with the R103 ring[25]. We applied DCPIB to the whole-cell recordings of PANX2 and found that application of 75 µM DCPIB inhibited the PANX2-mediated currents but showed no effects on PANX1-mediated currents (Fig. 5d–f). Thus, DCPIB inhibits both LRRC8A and PANX2 channels, and the shared extracellular Arg ring appears to be the major molecular and structural determinant of DCPIB blockade. This was further supported by mutagenesis studies in that a single point mutation at this position in PANX2 (R89A, R89W or R89E) rendered the mutant channel insensitive to application of DCPIB (Supplementary Fig. 6b, d).

## Discussion

This work now provides a glimpse into the structure, function, and pharmacology of human PANX2 channel. Cryo-EM structure determination was facilitated by fusion of a thermostabilized BRIL protein to the C-terminus of the PANX2 channel core. However, density

corresponding to the C-terminally fused BRIL protein was absent in the cryo-EM reconstruction, indicating that BRIL was connected to the PANX2 channel core through a flexible linker (residues 369–372). Consequently, BRIL does not interfere with channel function in that PANX2$_{EM}$ maintains the wild-type channel properties. This is in accordance with our previous cryo-EM analysis of the proton-activated chloride channel TMEM206, which was also made possible through a flexible C-terminal BRIL fusion[38]. In both scenarios, the BRIL fusion increased the expression level and biochemical stability of suboptimal membrane proteins such as TMEM206 and PANX2, thus aiding structure determination though BRIL itself was not rigidly fixed in the fusion constructs to augment particle alignment in cryo-EM image processing. Therefore, fusion with a flexibly attached stabilizing chaperone BRIL may represent a broadly applicable method to improve cryo-EM reconstruction of otherwise intractable membrane proteins. Furthermore, the 'flexible' BRIL fusion strategy complements the recently developed 'rigid' fusion approach[48], in which the membrane protein of interest is rigidly connected to BRIL. Subsequent application of synthetic antibodies against BRIL increases the size of the structurally ordered particles and thus enhances image alignment in single-particle cryo-EM analyses.

Application of DCPIB and analogs to selectively inhibit VRAC activity has served as an invaluable tool to investigate VRAC physiology[47,49,50]. Now we find that DCPIB also inhibits PANX2 channel activity and that PANX2 displays a similar anion permeability sequence as VRAC. Thus, the cross inhibition by DCPIB and the shared anion preference raise an important concern in functional studies of these channels in a cellular context by pharmacological manipulation. Physiological consequences of DCPIB blockade may stem from inhibition of PANX2 channel activity, rather than VRAC activity, or inhibition of both channels. Our structural and functional analysis of PANX2 represents an important step forward towards better understanding PANX2 channel physiology and facilitates the development of selective channel modulators.

## Methods

### Construct design and purification of PANX2

The codon-optimized DNA fragment encoding *H. sapiens* PANX2 (hPANX2, NCBI: NP_443071.2) was synthesized (Bio Basic Inc.) and cloned into a modified yeast expression vector pPICZ-B with a PreScission protease cleavage site followed by a C-terminal GFP-His$_{10}$ tag. Initial fluorescence-detection size-exclusion chromatography (FSEC) screening showed that the full-length wild-type channel had a low level of expression. A segment of 305 amino acids from the C-terminus was deleted to improve protein expression. For structural studies, thermostabilized apocytochrome b$_{562}$RIL (BRIL) was fused to the truncated construct to further improve protein expression and stability. The final expression construct PANX2$_{EM}$ included residues 1 to 372 of human PANX2 and a C-terminal BRIL followed by the PreScission protease cleavage site and GFP-His$_{10}$ tag. For electrophysiological recordings, the corresponding DNA fragments were ligated into a modifed pCEU vector containing a C-terminal GFP-His$_8$ tag. Point mutations in this study were generated by site-directed mutagenesis.

Yeast cells (*P. pastoris* strain SMD1163H, Invitrogen, #C17500) expressing PANX2$_{EM}$ were disrupted by milling (Retsch MM400) and resuspended in buffer containing 50 mM Tris pH 8.0 and 150 mM NaCl in the presence of protease inhibitors (2.5 µg ml$^{-1}$ leupeptin (L-010-100, GoldBio), 1 µg ml$^{-1}$ pepstatin A (P-020-100, GoldBio), 100 µg ml$^{-1}$ 4-(2-Aminoethyl) benzenesulfonyl fluoride hydrochloride (A-540-10, GoldBio), 3 µg ml$^{-1}$ aprotinin (A-655-100, GoldBio), 1 mM benzamidine (B-050-100, GoldBio) and 200 µM phenylmethane sulphonylfluoride (P-470-25, GoldBio)) and DNase I (D-300-1, GoldBio). The cell mixture was extracted with 1% (w/v) lauryl maltose neopentyl glycol (LMNG, NG310,

Anatrace) for 2 h with stirring at 4 °C and then centrifuged for 1 h at 30,000 × *g*. The supernatant was collected and incubated with 3 ml of cobalt-charged resin (786-403, G-Biosciences) for 3 h at 4 °C. Resin was then collected and washed with 10 column volumes of buffer containing 20 mM Tris pH 8.0, 150 mM NaCl, 20 mM imidazole and 85 µM glyco-diosgenin (GDN, GDN101, Anatrace). The protein was eluted with buffer containing an increased concentration of imidazole (200 mM), digested with PreScission protease at 4 °C overnight to remove the C-terminal GFP-His$_{10}$ tag, concentrated using a 100 kDa molecular weight cutoff concentrator, and further purified on a Superose 6 Increase 10/300 gel filtration column (GE Healthcare Life Sciences) pre-equilibrated in buffer containing 20 mM Tris pH 8.0, 150 mM NaCl, and 40 µM GDN. Peak fractions containing the channel protein were concentrated to ~6 mg ml$^{-1}$ and immediately used for cryo-EM grid preparations.

### Cryo-EM sample preparation and imaging

Prior to grid preparation, the purified protein was incubated with 1 mM Fluorinated Fos-Choline 8 (F300F, Anatrace) to improve particle orientation within the vitrified ice. A volume of 3.5 µl of purified channel protein at a concentration of ~6 mg/ml was pipetted onto glow-discharged copper Quantifoil R2/2 holey carbon grids (Q350CR2, Electron Microscopy Sciences), which had previously been cleaned for 1 min in an H$_2$/O$_2$ plasma using a Gatan Solarus 950 (Gatan). Grids were blotted for 2 s at ~100% humidity and then plunge frozen in liquid ethane using a Vitrobot Mark IV (ThermoFisher Scientific). The grids were then loaded into a Titan Krios G3 (ThermoFisher Scientific) cryo-electron microscope operating at an accelerating voltage of 300 kV with Falcon 4 Detector. Images were recorded with EPU software (ThermoFisher Scientific) in counting mode with a pixel size of 0.9 Å and a nominal defocus value ranging from −1.0 to −2.4 µm. Single-particle cryo-EM data were collected with a dose of ~5.23 electrons per Å$^2$ per second, and each movie was recorded with a 9.36 s exposure with 46 total frames (representing an accumulated dose of ~49 electrons/Å$^2$).

### Image processing and map calculation

Recorded movies were aligned and dose weighted with MotionCor2[51] and then subjected to contrast transfer function (CTF) determination using Gctf[52]. After motion correction and CTF estimation, low-quality images were manually removed from the data set. 20,264 particles were picked using LoG-based auto-picking to generate 2D classes in RELION3[53]. Template-based automatic picking resulted in 467,552 particles from 2847 micrographs. Particles were extracted using a box size of 280 pixels and subjected to 2D classification with a mask diameter of 180 Å. Low-quality particle images were discarded by performing two rounds of 2D classification. The resulted 2D classes containing 123,155 particles were selected and imported into cryoSPARC v3[54] to generate an initial map for 3D classification in RELION3. One of the 3D classes with 67,540 particles showing a complete channel was subjected to 3D refinement, reaching a nominal resolution of 4.66 Å. Subsequently, one round of 3D classification without particle alignment was performed. One of the 3D classes with 25,191 particles was selected and imported into cryoSPARC for nonuniform refinement, which resulted in an overall resolution of 3.92 Å.

### Model building and coordinate refinement

The AlphaFold[39] model of human PANX2 was used as an initial model, and subsequent manual adjustment was conducted in COOT[55]. Cycles of model building in COOT and real-space refinement using real_space_refine against the full map in PHENIX[56] were conducted to obtain the final refined atomic model, which was validated using MolProbity[57]. Pore radius calculation was performed using the program HOLE[58]. Structural figures were generated using UCSF ChimeraX[59].

## Electrophysiological recordings

The wild-type human PANX1, xenopus PANX1, human PANX2 channel, or each mutant channel was transfected by using Lipofectamine® 2000 reagent (11668019, Invitrogen) into HEK293T cells (Thermofisher, #R79007) for electrophysiological experiments. 24–36 h after transfection, whole-cell patch-clamp recordings were performed at room temperature using Axon 700B amplifier (Molecular Devices). The patch electrodes had a resistance of 3–5 MΩ and were prepared from borosilicate glass capillaries (BF 150-86-10, Sutter Instrument) using a Sutter P-1000 micropipette puller. The intracellular solution contained 147 mM NaCl, 10 mM EGTA, 10 mM HEPES pH 7.0 (pH was adjusted with NaOH) while the external solution contained 147 mM NaCl, 10 mM HEPES pH 7.4, 13 mM Glucose, 2 mM KCl, 2 mM $CaCl_2$, 1 mM $MgCl_2$. Whole-cell membrane currents were recorded using a voltage ramp from −100 to +100 mV for 2 s applied every 10 s with a holding potential at 0 mV. Currents were filtered at 2 kHz and digitized at 10 kHz. Clampex 10.4 software (Molecular Devices) was used for data acquisition and Clampfit 10 (Molecular Devices) was used for data analysis. Currents of PANX1, PANX2, or PANX2 mutants were normalized by cell capacitance to calculate current densities.

A multi-channel micro-perfusion system was used for pharmacological inhibition experiments using CBX (C4790, Sigma) and DCPIB (1540, TOCRIS). The xenopus and human PANX1 constructs were used for CBX and DCPIB experiments, respectively. For anion selectivity experiments, a 3 M KCl–agar bridge was used to reduce the drift of junction potentials. The bath solution was replaced with 10 mM HEPES pH 7.4, 2 mM Ca-Gluconate, 1 mM Mg-Gluconate, 15 mM Sucrose, and 147 mM NaX (NMDG-Cl), where X was $Cl^-$, $Br^-$, $I^-$, or Gluconate. The modified Goldman−Hodgkin−Katz equation was used to calculate the permeability ratios based on shifts in reversal potentials. For determining the Na/Cl permeability ratios with the Goldman−Hodgkin−Katz equation, we measured reversal potentials using a reduced salt bath solution containing 14.7 mM NaCl, 10 mM HEPES pH 7.4, 2 mM Ca-Gluconate, 1 mM Mg-Gluconate, and 245 mM sucrose.

## Immunofluorescence staining

The HEK293T cells were transfected with GFP-tagged human PANX2, PANX2$_{EM}$, or mutants using Lipofectamine® 2000 reagent (11668019, Invitrogen). 24 h after transfection, HEK293T cells were fixed with 4% paraformaldehyde for 30 min and washed two times with phosphate-buffered saline (PBS) buffer. Cells were then incubated with a wheat germ agglutinin (WGA; 1:400, 29023, Biotium,) lectin tagged with Biotium's CF 594 (5 g ml$^{-1}$ in Hanks' balanced salt solution without phenol red) for 10 min at 37 °C following the manufacturer's instruction. Cells were then mounted with 4′,6-diamidino-2-phenylindole (DAPI) Fluromount-G (0100-20, SouthernBiotech) after washing with PBS, and analyzed using a Nikon confocal microscope. For each channel protein, more than 25 cells from at least three different coverslips were selected for colocalization analysis. WGA was used to stain the cell membrane, and the expression levels of the WT PANX2 and mutants were measured by their GFP intensities. Specifically, the intensity of colocalized GFP was quantified and normalized by the total intensity of WGA. The normalized GFP intensity indicates the membrane expression level of the WT PANX2 or mutants. ImageJ software was used to determine the colocalization ratios of the GFP$^+$ area in the WGA$^+$ area, representing the expression levels on the plasma membrane. All the data were collected and analyzed with GraphPad prism 8.0.

## Data analysis

Data are shown as mean ± SEM. Statistical significances were determined using *t*-test (for two groups) or one-way ANOVA followed by a Tukey-Kramer post hoc test (for three or more groups). Difference between groups is considered statistically significant if $p < 0.05$.

## Reporting summary

Further information on research design is available in the Nature Portfolio Reporting Summary linked to this article.

## Data availability

The data that support this study are available from the corresponding authors upon request. The cryo-EM map has been deposited to Electron Microscopy Data Bank with accession code EMD-28902 (hPANX2). Atomic coordinates have been deposited to the Protein Data Bank (PDB) with accession code 8F7C (hPANX2). Previously published PDB codes that are referred to include 6UZY and 6NZW. The source data underlying Figs. 1c, 5c, and 5f, and Supplementary Fig. 4b, 4c, 5, and 6 are provided as a Source Data file. Source data are provided with this paper.

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

## Acknowledgements

This work was partly supported by NIH grants R01NS099341 (to P.Y.) and R01DK103901, R01AR077183, and R01AA027065 (to H.H.).

## Author contributions

Z.H conducted protein biochemistry and cryo-EM structure determination and analysis. Y.Z. performed electrophysiology experiments. Z.H., M.J.R., and J.A.J.F. collected cryo-EM images. R.S. provided reagents and cell lines. P.Y. and H.H. supervised the project. Z.H., Y.Z., H.H., and P.Y. wrote the manuscript. All authors edited the manuscript. Correspondence and requests for materials should be addressed to H.H. or P.Y.

## Competing interests

The authors declare no competing interests.
