## [Peer Review File · Nature Communications]

Structural and functional analysis of human pannexin 2 channelReviewers' Comments:

Reviewer #1:

Remarks to the Author:

The authors present near-atomic structure of the human PANX2 hemi-channel resolved using single-particle cryo-EM at 3.92 Å resolution. This structure revealed a heptameric architecture that resembles PANX1 and differs from previously assumptions of hexameric assembly. The authors also found, similar to PANX1, that the structurally disordered N-terminal amino acid residues are essential for PANX2 function, through currently unclear mechanism. By comparison with PANX1 and LRRC8A, the authors identified extracellular components that contribute to ion selectivity as well as pharmacology. This work presents important progress toward understanding the function and regulation of PANX2. I recommend publication of this paper with a few minor issues being addressed.

Minor points:

Fig.2 When showing amplified crevice, it would be nice to hide the subunits opposite to the pore (like in the overall view) so that it is easier to see lipids occupying the crevice.

Page 7. "two immediate possibilities. First, lipids may directly modulate channel activity such as blocking ion conduction by inserting their hydrophobic acyl chains into the central permeation path...". Please comment on whether conduction is known modulated by specific types of lipids. Is the larger pore diameter in the TM reasonable for blocking by lipids?

Page 9. "...each subunit of the PANX2 channel closely resembles that of PANX1... heptameric PANX1 and PANX2 channel assemblies deviate from each other". How do same building blocks assembly into different heptamers? What structural feature(s) in the monomer led to/determined the differences in assembly?

Page 10. "...extracellular SF suggest distinct ion selectivity and pharmacology of PANX2 from PANX1." Please comment on known data of selectivity and pharmacology and how it correlates with respective SF structures.

Page 11. "...Furthermore, PANX2 was permeable to large anions such as gluconate but not to large cations such as N-methyl-D-glucamine ...". This contradicts Sup Fig 4b where the reversal of NMDG-Cl is at 0 (same as NaCl), which suggests the same conductance of Na⁺ and NMDG. If NMDG is impermeant, the reversal should be around -10mV given a 2:1 Cl:Na permeability (~150mM NaCl inside and 150mM NMDG-Cl outside).

Page 12. "...though surface expression of these mutant channels is comparable to that of the wild type (Supplementary Fig. 4c, 4d)..." Sup. Fig. 4c is unclear. Was the intensity of GFP at plasma membrane quantified or the colocalization ratios? Colocalization ratios do not contain intensity information and thus level of expression.

Sup. Fig. 5c, d, e, f: Coloring is a bit confusing. Are PANX2/R89A/R89W/R89E recorded in different ionic conditions (NaCl, NaBr, NaI, Na-Gluconate) or only in symmetrical NaCl (or low outside NaCl)?

Sup. Fig.5f: supposedly these traces are recorded in symmetrical NaCl? The reversals in panel f seem the same (~0) for all R89 mutants, but around -10mV for WT that is likely due to limited accuracy arising from noise and leak (for some mutants I/V is not significantly different from the non-transfected control shown in Fig. 1b). It would be nice to show the I/V traces of low outside NaCl that are used for the calculation of permeability ratios (in panel e) for assessment of measurement reliability.

Fig. 5: Are the effects CBX (0.1mM) and DCPIB (0.075mM) reversible? Are the DCPIB affinities/IC50 similar or different between PANX2 and LRRC8A?

Fig. 1b. PANX2 seems to show outward rectification. Is this a result of voltage-dependent gating or channel blocking etc?

Figure panel labels (a, b, c, ...): Are the letters compressed vertically?

Page 9. Unstructured N-terminus being essential for channel function is very interesting. This would be a topic for further research, but it would still be nice to briefly discuss possible underlying mechanisms.

Reviewer #2:

Remarks to the Author:

Pannexin ion channels are essential for many physiological processes. ATP-releasing PANX1 controls purinergic signaling, and PANX2 play important roles in skin and neuronal cells. This family of ion channels have attracted tremendous attention in the field. Substantial progress has been made to understand the mechanism of PANX1. However, our understanding of PANX2 and PANX3 is still limited. In this manuscript, the authors reported the first structure of PANX2 and systematic functional characterizations, which provide novel insights into this important family of ion channels. Unexpectedly, the authors showed that PANX2 is closely related to LRRC8A (a volume-regulated anion channel) but distinct from PANX1 in terms of selectivity filter structure, ion selectivity, and pharmacology. These findings not only lay the foundation for understanding physiological functions of pannexin channels but also establish an intriguing link between different families of channels. The finding that PANX2 can be inhibited by a potent inhibitor for VRAC channel also calls for further investigation of VRAC and PANX2 physiological functions. Overall, this manuscript is of high quality, and findings are novel and significant. It makes a very important contribution to the field and represents a significant advance.

Several points need to be addressed:

1) R89 forms the selectivity filter in PANX2. What is the quality of R89 side chain density? It would be helpful to show the local density map.

It appears that the positively charged R89 side chains are relatively close to each other in the pore. Is this an energetically stable state? Are there any ion densities at the center of the selectivity filter?

2) The R89E mutant still shows anion selectivity over cation. Are there any other molecular determinants that may contribute to the ion selectivity besides the selectivity filter?

The electrostatic potential of the intracellular cavity of PANX2 looks quite different from that of PANX1 and LRRC8A. Does this contribute to its substrate selectivity or conduction?

3) It is not clear why removing N-terminal unstructured 40 aa gives rise to a non-functional channel given that the pore presumably should remain open. Any hypothesis on this?

4) Do helix dipoles contribute to the ion conduction/selectivity of PANX2? It appears that the ends of helices point at the center of the pore.

Minor:

p.8, 2nd paragraph: 'positioning the N-terminal ends of seven E1H helices to line the extracellular entrance'. It would be helpful to clarify the seven E1H helices are from seven protomers.

REVIEWERS' COMMENTS

Reviewer #1 (Remarks to the Author):

The authors present near-atomic structure of the human PANX2 hemi-channel resolved using single-particle cryo-EM at 3.92 Å resolution. This structure revealed a heptameric architecture that resembles PANX1 and differs from previously assumptions of hexameric assembly. The authors also found, similar to PANX1, that the structurally disordered N-terminal amino acid residues are essential for PANX2 function, through currently unclear mechanism. By comparison with PANX1 and LRRC8A, the authors identified extracellular components that contribute to ion selectivity as well as pharmacology. This work presents important progress toward understanding the function and regulation of PANX2. I recommend publication of this paper with a few minor issues being addressed.

Minor points:

Point 1: Fig.2 When showing amplified crevice, it would be nice to hide the subunits opposite to the pore (like in the overall view) so that it is easier to see lipids occupying the crevice.

Response: We thank the reviewer for this good point. We have now removed the subunits opposite to the pore, similar to the overall view, in the new Fig. 2.

Point 2: Page 7. “two immediate possibilities. First, lipids may directly modulate channel activity such as blocking ion conduction by inserting their hydrophobic acyl chains into the central permeation path...”. Please comment on whether conduction is known modulated by specific types of lipids. Is the larger pore diameter in the TM reasonable for blocking by lipids?

Response: Specific lipids that modulate PANX2 (or PANX1) conduction have not been reported. The reviewer raises a very good question. The large pore diameter in the TM is reasonable for blocking by lipids, and these have been demonstrated by recent cryo-EM structures of PANX1 and CALHM channels. To make these points clear, we have now stated the following on Page 8 in the main text.

“Notably, how lipids modulate PANX2 channel activity remains to be investigated. Nonetheless, the large pore dimension in the transmembrane region would allow pore-residing lipids block conduction, as suggested by cryo-EM structures of related large-pore channels including PANX1 and CALHM channels^{14,27}.”

Point 3: Page 9. “...each subunit of the PANX2 channel closely resembles that of PANX1... heptameric PANX1 and PANX2 channel assemblies deviate from each other”. How do same building blocks assembly into different heptamers? What structural feature(s) in the monomer led to/determined the differences in assembly?

Response: We meant that the PANX1 and PANX2 heptamers deviate from each other to a larger extent compared with single channel subunits (an RMSD of ~1.9 Å for Ca atoms in TM1-TM4 in a single subunit vs an RMSD of ~3.7 Å in the heptameric channel). This is partially due to very low sequence identity (27%) between PANX1 and PANX2 proteins. Differences in subunit structure and inter-subunit packing both contribute to a larger deviation in heptameric channels.

Point 4: Page 10. "...extracellular SF suggest distinct ion selectivity and pharmacology of PANX2 from PANX1." Please comment on known data of selectivity and pharmacology and how it correlates with respective SF structures.

Response: PANX2 selectivity and pharmacology have not been studied until this work. We have detailed results and discussion of our studies of PANX2 selectivity and pharmacology and comparison with PANX1 and LRRC8 channels in later sections in this manuscript.

Point 5: Page 11. "...Furthermore, PANX2 was permeable to large anions such as gluconate but not to large cations such as N-methyl-D-glucamine ...". This contradicts Sup Fig 4b where the reversal of NMDG-Cl is at 0 (same as NaCl), which suggests the same conductance of Na⁺ and NMDG. If NMDG is impermeant, the reversal should be around -10mV given a 2:1 Cl:Na permeability (~150mM NaCl inside and 150mM NMDG-Cl outside).

Response: We thank the reviewer for this correction. We have now stated the following on Page 12.

"...PANX2 was permeable to large anions and cations such as gluconate and N-methyl-D-glucamine."

Point 6: Page 12. "...though surface expression of these mutant channels is comparable to that of the wild type (Supplementary Fig. 4c, 4d)..." Sup. Fig. 4c is unclear. Was the intensity of GFP at plasma membrane quantified or the colocalization ratios? Colocalization ratios do not contain intensity information and thus level of expression.

Response: We appreciate the comments and have now made it clear by including more details in the revised *Methods*. We used WGA to stain the cell membrane, and the expression levels of the WT PANX2 and mutants were estimated by the GFP intensity. The intensity of colocalized GFP was quantified and normalized by the total intensity of WGA. The normalized intensity indicates the membrane expression levels of the WT PANX2 or mutant channels. We have now included these details in the revised *Methods*.

"WGA was used to stain the cell membrane, and the expression levels of the WT PANX2 and mutants were measured by their GFP intensities. Specifically, the intensity of colocalized GFP was quantified and normalized by the total intensity of WGA. The normalized GFP intensity indicates the membrane expression level of the WT PANX2 or mutants."

Point 7: Sup. Fig. 5c, d, e, f: Coloring is a bit confusing. Are PANX2/R89A/R89W/R89E recorded in different ionic conditions (NaCl, NaBr, NaI, Na-Gluconate) or only in symmetrical NaCl (or low outside NaCl)?

Response: We appreciate the comments. To make it clear, we have now improved the labeling in the figure, and also included detailed recording conditions in the figure legend. In Sup. Fig. 5d, currents for PANX2/R89A/R89W/R89E were recorded in symmetrical NaCl conditions (External solution comprises 147 mM NaCl, 10 mM HEPES pH 7.4, 13 mM Glucose, 2 mM KCl, 2 mM CaCl₂, 1 mM MgCl₂ and the internal solution contains 147 mM NaCl, 10 mM EGTA, 10 mM HEPES pH 7.4). In Sup. Fig. 5e, PANX2/R89A/R89W/R89E were recorded with low NaCl external solution containing 14.7 mM NaCl, 10 mM HEPES pH 7.4, 2 mM Ca-Gluconate, 1 mM Mg-Gluconate, and 245 mM sucrose. The internal solution contained 147 mM NaCl, 10 mM EGTA, 10 mM HEPES pH 7.4.

Point 8: Sup. Fig.5f: supposedly these traces are recorded in symmetrical NaCl? The reversals in panel f seem the same (~0) for all R89 mutants, but around -10mV for WT that is likely due to limited accuracy arising from noise and leak (for some mutants I/V is not significantly different from the non-transfected control shown in Fig. 1b). It would be nice to show the I/V traces of low outside NaCl that are used for the calculation of permeability ratios (in panel e) for assessment of measurement reliability.

Response: We thank the reviewer for this thoughtful comment. The current traces in Sup.Fig. 5f were recorded in symmetrical NaCl (external solution contains 147 mM NaCl, 10 mM HEPES pH 7.4, 13 mM Glucose, 2 mM KCl, 2 mM CaCl₂, 1 mM MgCl₂, and the internal solution contains 147 mM NaCl, 10 mM EGTA, 10 mM HEPES pH 7.4). The typical reversals were around 0 mV for our recordings. We have now replaced the traces with more representable ones in Sup. Fig 5f. As suggested, we have now also added additional traces used for the calculation of permeability ratios (in panel e) with external solutions containing 147 mM NaI and 14.7 mM NaCl. The following recording conditions were used: External solutions containing 147 mM NaCl or NaI, 10 mM HEPES pH 7.4, 2 mM Ca-Gluconate, 1 mM Mg-Gluconate, and 15 mM Sucrose were used for recording currents with 147 mM outside NaCl or NaI, respectively. External solution containing 14.7 mM NaCl, 10 mM HEPES pH 7.4, 2 mM Ca-Gluconate, 1 mM Mg-Gluconate, and 245 mM sucrose was used for recording currents with 14.7 mM outside NaCl. The internal solution comprised 147 mM NaCl, 10 mM EGTA, 10 mM HEPES pH 7.4 for all recordings.

Point 9: Fig. 5: Are the effects CBX (0.1mM) and DCPIB (0.075mM) reversible? Are the DCPIB affinities/IC₅₀ similar or different between PANX2 and LRRC8A?

Response: We appreciate this point. Although 0.1 mM CBX reversibly suppressed PANX1 currents, it had no effect on PANX2. Moreover, 75 μ M DCPIB could suppress around 50% of the PANX2 current in a reversible manner without significant inhibition on PANX1. Gunasekar et al., 2022 reported that the IC₅₀ of DCPIB for LRRC8A is about 3.9 μ M. Therefore, DCPIB has a higher affinity for LRRC8A than PANX2.

Point 10: Fig. 1b. PANX2 seems to show outward rectification. Is this a result of voltage-dependent gating or channel blocking etc?
Figure panel labels (a, b, c, ...): Are the letters compressed vertically?

Response: PANX2 is constitutively active with outward rectification in our recording conditions. The molecular mechanisms underlying outward rectification would need to be further investigated. Panel labels are now fixed.

Point 11: Page 9. Unstructured N-terminus being essential for channel function is very interesting. This would be a topic for further research, but it would still be nice to briefly discuss possible underlying mechanisms.

Response: We agree with the reviewer that the unstructured N-terminus would be a topic for further research. We have discussed the studies of the unstructured but essential N-terminus in PANX1, which suggest that the N-terminus is involved in rearrangement of pore-blocking lipids that is obligatory for channel opening. Because we did not observe lipid densities inside the pore lumen of PANX2 and we have implied the similarities between PANX2 and PANX1 N-termini, we would like to speculate no further.

Reviewer #2 (Remarks to the Author):

Pannexin ion channels are essential for many physiological processes. ATP-releasing PANX1 controls purinergic signaling, and PANX2 play important roles in skin and neuronal cells. This family of ion channels have attracted tremendous attention in the field. Substantial progress has been made to understand the mechanism of PANX1. However, our understanding of PANX2 and PANX3 is still limited. In this manuscript, the authors reported the first structure of PANX2 and systematic functional characterizations, which provide novel insights into this important family of ion channels. Unexpectedly, the authors showed that PANX2 is closely related to LRRC8A (a volume-regulated anion channel) but distinct from PANX1 in terms of selectivity filter structure, ion selectivity, and pharmacology. These findings not only lay the foundation for understanding physiological functions of pannexin channels but also establish an intriguing link between different families of channels. The finding that PANX2 can be inhibited by a potent inhibitor for VRAC channel also calls for further investigation of VRAC and PANX2 physiological functions. Overall, this manuscript is of high quality, and findings are novel and significant. It makes a very important contribution to the field and represents a significant advance.

Several points need to be addressed:

Point 1: R89 forms the selectivity filter in PANX2. What is the quality of R89 side chain density? It would be helpful to show the local density map.

It appears that the positively charged R89 side chains are relatively close to each other in the pore. Is this an energetically stable state? Are there any ion densities at the center of the selectivity filter?

Response: The R89 side chain density is well resolved. We have now included cryo-EM density of this region in the new Supplementary Fig. 2. We did not observe strong ion densities at the center of the selectivity filter. For comparison, the Arg side chains are even closer in the reported LRRC8A structure and no ion densities were observed at the center (References: Deneka, D., Sawicka, M., Lam, A. K. M., Paulino, C. & Dutzler, R. Structure of a volume-regulated anion channel of the LRRC8 family. *Nature* 558, 254–259, 2018). The diameter of the Arg ring in our structure is wide enough to accommodate solvent ions such as Cl⁻. Perhaps these ions are not ordered enough to be resolved by cryoEM.

Point 2: The R89E mutant still shows anion selectivity over cation. Are there any other molecular determinants that may contribute to the ion selectivity besides the selectivity filter?

The electrostatic potential of the intracellular cavity of PANX2 looks quite different from that of PANX1 and LRRC8A. Does this contribute to its substrate selectivity or conduction?

Response: The reviewer is correct that R89E still shows weak anion selectivity and that the electrostatic potential of the lumen of PANX2 is quite different from those in PANX1 and LRRC8A. The electrostatic potential likely contributes to differing permeation ratios of anion over cation in these channels.

Point 3: It is not clear why removing N-terminal unstructured 40 aa gives rise to a non-functional channel given that the pore presumably should remain open. Any hypothesis on this?

Response: We agree with the reviewer that it is not clear why removing the N-terminus gives rise to a non-functional channel. We have discussed the studies of the analogous unstructured

but essential N-terminus in PANX1, which suggest that the N-terminus is involved in rearrangement of pore-blocking lipids that is obligatory for channel opening. We did not observe lipid densities inside the pore lumen of PANX2. However, in analogy, perhaps the N-terminus plays a similar role in PANX2 as in PANX1.

Point 4: Do helix dipoles contribute to the ion conduction/selectivity of PANX2? It appears that the ends of helices point at the center of the pore.

Response: This is a good point. Contribution of the helix dipoles has been proposed by earlier studies in LRRC8A. We have now included citation of this work and included the following on page 8 in our revised main text.

“...the seven arginine amino acids form a basic ring structure generating an electropositive extracellular constriction, which is additionally contributed by the E1H helix dipole²⁵.”

Minor:

Point 5: p.8, 2nd paragraph: ‘positioning the N-terminal ends of seven E1H helices to line the extracellular entrance’. It would be helpful to clarify the seven E1H helices are from seven protomers.

Response: We thank the reviewer for this point and have now revised this sentence.

“...positioning the N-terminal ends of seven E1H helices from the heptameric channel to line the extracellular entrance...”